# Non-Invasive Evaluation of Cerebral Microvasculature Using Pre-Clinical MRI: Principles, Advantages and Limitations

**DOI:** 10.3390/diagnostics11060926

**Published:** 2021-05-21

**Authors:** Bram Callewaert, Elizabeth A. V. Jones, Uwe Himmelreich, Willy Gsell

**Affiliations:** 1Biomedical MRI Group, University of Leuven, Herestraat 49, bus 505, 3000 Leuven, Belgium; bram.callewaert@kuleuven.be (B.C.); willy.gsell@kuleuven.be (W.G.); 2CMVB, Center for Molecular and Vascular Biology, University of Leuven, Herestraat 49, bus 911, 3000 Leuven, Belgium; liz.jones@kuleuven.be; 3CARIM, Maastricht University, Universiteitssingel 50, 6200 MD Maastricht, The Netherlands

**Keywords:** microvasculature, brain, MRI, rodent, neurodegenerative disorders, perfusion

## Abstract

Alterations to the cerebral microcirculation have been recognized to play a crucial role in the development of neurodegenerative disorders. However, the exact role of the microvascular alterations in the pathophysiological mechanisms often remains poorly understood. The early detection of changes in microcirculation and cerebral blood flow (CBF) can be used to get a better understanding of underlying disease mechanisms. This could be an important step towards the development of new treatment approaches. Animal models allow for the study of the disease mechanism at several stages of development, before the onset of clinical symptoms, and the verification with invasive imaging techniques. Specifically, pre-clinical magnetic resonance imaging (MRI) is an important tool for the development and validation of MRI sequences under clinically relevant conditions. This article reviews MRI strategies providing indirect non-invasive measurements of microvascular changes in the rodent brain that can be used for early detection and characterization of neurodegenerative disorders. The perfusion MRI techniques: Dynamic Contrast Enhanced (DCE), Dynamic Susceptibility Contrast Enhanced (DSC) and Arterial Spin Labeling (ASL), will be discussed, followed by less established imaging strategies used to analyze the cerebral microcirculation: Intravoxel Incoherent Motion (IVIM), Vascular Space Occupancy (VASO), Steady-State Susceptibility Contrast (SSC), Vessel size imaging, SAGE-based DSC, Phase Contrast Flow (PC) Quantitative Susceptibility Mapping (QSM) and quantitative Blood-Oxygenation-Level-Dependent (qBOLD). We will emphasize the advantages and limitations of each strategy, in particular on applications for high-field MRI in the rodent’s brain.

## 1. Introduction

Microvascular health underlies the physiology of all organs. Capillaries have a key role in the exchange of oxygen, carbon dioxide, nutrients and hormones to all cells around the body. Microcirculatory impairments can result in altered microvascular perfusion and tissue oxygenation which can lead to tissue damage and organ dysfunction. Due to the brain’s high energy requirements and its inability to store energy, an adequate cerebral perfusion, maintaining oxygen and nutrients homeostasis, is extremely important for normal functioning. Alterations to the cerebral microcirculation have been recognized to play a crucial role in the development of several neurodegenerative disorders such as vascular cognitive impairment [1,2,3,4,5], Alzheimer’s disease [6,7,8,9,10], Parkinson’s disease [11,12,13] and Huntington’s disease [14,15,16]. However, the exact role of the microvascular alterations in the pathophysiological mechanisms leading to neurodegenerative disorders often remains poorly understood. In addition, microvascular dysfunction is thought to play a role in neurological disorders such as stroke [17] and psychiatric disorders such as schizophrenia [18,19] and autism spectrum disorder [20].

There is increasing evidence that endothelial dysfunction is one of the earliest events in the initiation of neurodegenerative disorders [21,22,23]. Endothelial dysfunction can lead to vascular inflammation, disruption of the vascular tone, impaired vasodilation and vasoconstriction, breakdown of the Blood Brain Barrier (BBB), vessel rarefaction and thrombosis [4,24,25,26,27,28,29]. This leads to reduced tissue perfusion and the disruption of the delivery of nutrients and oxygen, resulting in ischemia and ultimately leading to damage or death of brain cells [30,31].

Microvessels are below the currently achievable spatial resolution of MRI and other clinically applicable imaging methods, making it impossible to perform direct non-invasive measurement of the microvascular density. However, MRI can be used to measure perfusion, providing indirect read-outs of the underlying microvasculature alteration. Due to its wide availability, the lack of ionizing radiation and its high spatial and temporal resolution, MRI has become one of the best-suited, non-invasive techniques to study the cerebral microcirculation and perfusion. The use of several hemodynamic parameters such as Cerebral Blood Volume (CBV), Cerebral Blood Flow (CBF), Mean Transit Time (MTT), vascular barrier permeability and tissue oxygenation provides valuable insights into the cerebral microvascular function, integrity and architecture. Therefore, MRI can be used as a promising tool to provide early detection and characterization of endothelial cell function and microvascular perfusion.

Although patient studies are useful, they often fail to link indirect imaging readouts with molecular and cellular processes, thus not providing sufficient information on the mechanism of disease development. Pre-clinical animal models can validate the non-invasive imaging with invasive techniques such as histology and immunohistochemistry that assess microvascular dysfunction and density [32,33,34]. Biopsies from patients can only rarely be obtained [35]. Post-mortem samples represent end points in neurodegenerative disease where the disease in patients is progressed to a state where conclusions on the onset and disease progression can hardly be drawn. Animal models allow the study of the disease mechanism at several stages of the development, as well as before the onset of symptoms, allowing investigation of the disease development [36]. Lastly, by using animal models, a high group homogeneity can be achieved which increases the reproducibility and statistical power of the experiments compared to clinical studies [37]. This makes pre-clinical MRI an important tool for the development and validation of novel MRI sequences for later applications in the clinic.

Rodents are by far the most common animals in pre-clinical research. While rodent imaging can be performed on human MRI scanners, it is highly beneficial to perform it on a high-resolution dedicated-small-animal MRI system [38]. In recent years, technical improvements have led to the increased availability of high-field small-animal MRI systems that achieve high spatial resolution in combination with excellent soft tissue contrast and a high signal-to-noise ratio (SNR) [39]. These improvements have resulted in a rapid growth in the use of MRI in rodents over the past years. However, pre-clinical MRI still has a number of technical and biological limitations.

This article reviews the cerebral perfusion MRI techniques: Dynamic Contrast Enhanced (DCE), Dynamic Susceptibility Contrast Enhanced (DSC) and Arterial Spin Labeling (ASL), followed by several less established imaging strategies used to analyze cerebral microcirculation. An overview of the MRI techniques described in this article can be found in Table 1. For each technique, we will discuss the basic principle and its microcirculatory parameters. We will emphasize the advantages and limitations of each strategy, in particular on applications for high-field small-animal MRI.

## 2. High Field Pre-Clinical MRI

The main advantage of dedicated-small-animal MRI systems is the combination of a high magnetic field strength and high gradient strength, resulting in a high SNR and resolution. This can be further increased by the use of dedicated coils that can be placed close to the animal, the use of cryogen cooled coils [40] and longer acquisitions [37,41]. However, the high magnetic field also causes several artifacts and limitations such as increased field inhomogeneity, increased specific absorption rate, susceptibility artifacts, chemical shift artifacts and changes in the relaxation times [37,42]. Pre-clinical MRI systems also often lack the implementation of pulse sequences that are used in well-established and optimized clinical protocols. The lack of standardization and technical limitations has led to great diversity across sequences and acquisition parameters, limiting the use and translatability to the clinic [43]. The standardization of these protocols and their parameters is crucial for an improved translatability.

Due to the smaller size of rodents and the fact that they would move during an MRI acquisition, the use of anesthesia is required. This has an influence on several body systems and can cause physiological alterations, which can influence the outcome of the imaging experiment [44]. To maintain stable physiological and hemodynamic conditions of anesthetized animals, close monitoring of the temperature, breathing rate and cardiac cycle of the animal under anesthesia is required [42]. Furthermore, the breathing rate, heart rate and velocity of blood flow are much higher in rodents compared to those of humans, requiring a high temporal resolution.

## 3. Cerebral Perfusion MRI Techniques

Blood perfusion is defined as the delivery of blood through the microvascular network to a tissue or organ. The main parameter used to assess perfusion in the brain is the CBF, which is expressed in volume of blood per unit of time per unit of tissue mass (mL/min/g). However, there are several additional physiological parameters such as blood volume, blood velocity and blood oxygenation that provide valuable information about the perfusion of tissues [45]. Quantitative perfusion weighted imaging allows the measurement and examination of perfusion maps of several hemodynamic parameters such as CBV, CBF, MTT, vessel permeability and tissue oxygenation, providing valuable insights into the microvascular function, integrity and architecture.

In general, perfusion MRI techniques can be divided into two categories: exogeneous techniques that make use of the injection of a contrast agent and the completely non-invasive endogenous contrast techniques (Table 1). The exogenous techniques result in a higher spatial resolution and are therefore more widely used [46]. The endogenous techniques are not dependent on the injection of a contrast agent, making it safer to study disease progression [45]. Furthermore, exogenous measurements are often limited to a single injection of the contrast agent, making it more difficult to perform repeated measurements during the same imaging session.

### 3.1. Dynamic Contrast Enhanced MRI

DCE-MRI is an exogenous contrast-based technique, mainly used to provide an estimation of the perfusion and the permeability of the cerebral microvasculature [47]. Using DCE-MRI, T1 changes of tissue over time, following the introduction of a paramagnetic or superparamagnetic contrast agent, can be measured. For paramagnetic contrast agents, the acquisition consists of a baseline image (T1_0_ map), followed by a series of T1-weighted images after an intravenous bolus of contrast agent. After injection, the contrast agent spreads through the tissue resulting in a change in the MR signal intensity proportional to the concentration of the contrast agent (T1 shortening). These temporal changes provide a signal intensity curve from which physiological parameters of the microvascular system such as perfusion, vessel permeability and the extravascular-extracellular space of the tissue can be derived [45,48].

There are two main approaches to analyze DCE measurements quantitatively, parametric and nonparametric approaches. The nonparametric (model free) or semi-quantitative techniques measure empirical metrics directly from the signal intensity curve such as the Bolus Arrival Time, Time-To-Peak, the maximal signal intensity or the area under the signal Attenuation Curve. If the MR signal is converted to a signal concentration curve, the latter is referred to as the initial area under the gadolinium curve. These metrics provide information about the shape and the structure of the MR signal intensity and can often be correlated to the underlying physiology [48,49,50]. Nonparametric approaches are suitable for fast and simple diagnostics and have been used in various (pre-)clinical studies. However, nonparametric techniques have some important limitations. They are sensitive to several parameters, such as the MRI acquisition protocol (sequence and parameters), type of scanner, injection protocol and type of contrast agent, making comparison of the results difficult [50]. Furthermore, the contribution of the physiological parameters to the MR signal intensity remains unclear [51].

Parametric or quantitative studies estimate the physiological parameters that can be directly related to the physiological properties [48]. The parametric evaluation of the signal intensity attenuation curve is done by converting the MR signal intensity to a time contrast concentration curve and fitting it to a model. In order to perform this kinetic modeling, the Arterial Input Function (AIF), describing the concentration changes in a blood vessel entering the tissue as a function over time, has to be identified. The accurate identification of the AIF is far from straightforward. A detailed description of the determination of the AIF can be found elsewhere [52].

Most of the studies use the two-compartmental pharmacokinetic (PK) model comprising the intravascular-extracellular space and the extravascular-extracellular space introduced by Tofts et al. (Figure 1) [53]. The two-compartment model estimates the volume transfer constant between the blood plasma and the extravascular-extracellular space (K^trans^); the reflux exchange rate between the extravascular-extracellular space and blood plasma (k_ep_) and the volume of the EES per unit volume of tissue (ν_e_ = K^trans^/k_ep_) (Table 2) [45,48,49].

The transfer constant K^trans^ is the most widely used kinetic parameter. It is a complex function depending on the blood flow, the vascular surface area per unit mass of tissue (representing the size and number of blood vessels) and the microvascular permeability, making it dependent on the tissue and its physiology [45,48]. The dependence of K^trans^ on both perfusion and microvascular permeability complicates the interpretation of the results, which is a major limiting factor of DCE-MRI. More complex DCE models, to describe the underlying tissue physiology more accurately, have been proposed [54]. However, the higher degree of complexity of these models, and the increased number of parameters to be estimated, often results in increased errors in the parameters estimation [48,55].

DCE-MRI is frequently used in both clinical and pre-clinical studies. Parametric DCE-MRI requires the fast and accurate determination of the T1 values. In practice, a trade-off must often be made between the temporal resolution, spatial resolution, the SNR and the field of view [55]. The limited spatial resolution in combination with increased susceptibility artifacts at higher field strength and fast changes in the blood plasma contrast agent often result in difficulties in the identification of the AIF in rodents [55,56]. In pre-clinical studies, it is especially important to optimize the temporal resolution, allowing accurate T1 determination, without reducing the spatial resolution. To decrease the relatively long acquisition times in MRI, several acceleration techniques have been proposed [57]. The use and implementation of acceleration techniques with high-field small-animal MRI systems has remained limited so far. Better implementations and increased use of acceleration techniques on high-field small-animal MRI scanners are needed to increase the reliability and reproducibility of the quantitative kinetic parameters in future pre-clinical DCE-MRI studies.

Since it is especially challenging to obtain a good identification of the AIF in small animals, several alternative approaches and models have been proposed. The AIF can also be determined by sampling arterial blood. However, due to their small blood volume, repeated blood sampling can have a significant effect on the physiological status of the animal [58]. Furthermore, the low sampling rate and the inability to collect blood samples close to the tissue of interest often confound the identification of the AIF [55,59]. An average population-based AIF is often used to avoid the need for a subject-specific AIF [60], but this approach does not take inter-subject variability into account.

Completely non-invasive methods have been developed to determine the AIF on an individual basis from the DCE-MRI [61,62]. These methods need high temporal resolution and require often the presence of a large vessel within the field of view. A potentially interesting alternative approach, which compares the obtained curve with that of a reference region in healthy tissue, is the reference region model [60]. This approach is frequently used in rodents and allows for quantitative analysis without the need of an AIF. Since this technique does not require a high temporal resolution to determine the AIF, a higher spatial resolution and/or SNR can be achieved. This has mainly been used in oncology to obtain quantitative measurements of perfusion in brain tumors [63,64].

Additional problems can arise due to the difficulty and lack of standardization of the contrast agent injection in rodents [65]. The administration of the contrast agent is typically performed through injection in the jugular vein or tail vein, which is often challenging. Changes in the bolus of the contrast agent can have a significant effect on the quantitative parameters [55]. Therefore, an automated injection system is preferred to increase the reproducibility compared to a manual injection.

### 3.2. Dynamic Susceptibility Contrast Enhanced MRI

The DSC MRI technique, also known as bolus tracking, is the most widely used MRI technique to measure brain perfusion [45,66]. Similar to DCE MRI, the acquisition consists of a pre-contrast baseline image followed by a series of MR images after an intravenous bolus of contrast agent. In DSC MRI, the tissue perfusion is assessed by evaluation of a series of rapidly repeated T2- or T2*-weighted MR images resulting from the first pass of a contrast agent bolus through the tissue [67,68]. During the first pass, the contrast agent is mainly confined to the intravascular space allowing a good estimation of the perfusion. The susceptibility induced signal loss over time (T2/T2* shortening), proportional to the contrast agent concentration, provides a signal intensity attenuation curve. The DSC-MRI technique is straightforward, has short acquisition times and has a high contrast to noise ratio compared to other perfusion methods. Its main limitation is the need for the identification of the AIF for quantitative analysis. In rodents, the capillary blood flow is about five times higher than in humans. Therefore, first pass measurements in rodents require extremely short bolus injections and acquisition times, making the MR signals much more sensitive to T1 changes and susceptibility artifacts [69].

Similar to DCE, the signal intensity attenuation curve can be used to derive semi-quantitative parameters. The transformation into the contrast concentration curve allows quantitative evaluation of the following physiological parameters: CBV, defined as the fraction of tissue volume occupied by blood (ml/g); CBF (ml/g/min) and MTT defined as the average time the contrast agent travels through the vasculature of the brain tissue (s) (Table 2). Figure 2 shows an example of a typical tissue contrast concentration curve and explains how it can be used to derive the physiological parameters. The absolute quantification of the physiological parameters is strongly dependent on the AIF, and most of the artifacts in the quantification can be linked directly to an incorrect identification of the AIF [52]. Since absolute quantification is difficult, the relative parameters calculated from the contrast concentration curve without the identification of the AIF are often used instead.

The DSC MRI technique relies on the assumption that the contrast agent remains intravascular. As the contribution of the permeability and the volume of the extravascular-extracellular space increases with the time after the bolus injection, DSC MRI requires a high temporal resolution. The disruption of the BBB can introduce errors in the measurements. In order to correct for the leakage of the contrast agent, a preload of contrast agent can be administered [70], or a leakage-correction algorithm can be used [70,71,72]. Recently, double or multi-echo approaches to remove the T1 effects caused by leakage of the contrast agent have gained increasing interest [73,74]. In addition to the leakage-correction, multi-echo DSC MRI was shown to improve identification of the AIF [74], estimation of the transfer constant K^trans^ and the volume of the extravascular-extracellular space per unit volume of tissue ν_e_ [75]. The multi-echo approach thus allows the extraction of the kinetic parameters and the conventional DSC parameters without the need for multiple contrast agent injections (Table 2) [75].

Both spin-echo (SE, T2) and gradient-echo (GE, T2*) sequences can be used for DSC MRI. The susceptibility contrast in GE images arises from the contributions of both macro-and microvasculature [76,77]. Therefore, it is preferable to use SE images, where the signal loss is mainly sensitive to the microvasculature [78]. Recently, combination of the two techniques referred to as Spin- and Gradient-Echo Echo-Planar Imaging (SAGE EPI) was proposed by Schmiedeskamp et al. [79].

### 3.3. Arterial Spin Labeling

ASL is the most frequently used, completely non-invasive, cerebral perfusion MRI technique (Table 1). It is based on the labeling, often referred to as ‘tagging’, of water spins in the arterial blood supply. The labeled blood water spins are used as an endogenous contrast agent. Several techniques combining different preparations and readout schemes exist for ASL. The general ASL scheme consists of two consecutive acquisitions. Before the first acquisition, the arterial blood water spins are tagged ‘upstream’ from the tissue of interest using an inversion pulse. After a delay time (Inversion Time, TI), during which the tagged spins travel to the tissue of interest and exchange with the stationary spins in the extravascular-extracellular space, an image is acquired (Figure 3A). The second acquisition, which serves as a control, is identical to the first acquisition without the tagging of the arterial water spins [80]. The difference between both images provides a signal proportional to the exchanged water magnetization and therefore the arterial blood flow to the tissue after the delay time. This difference is therefore directly proportional to the capillary blood flow in the tissue [81]. The signal intensities from the perfusion weighted images can then be converted to quantitative measurements of CBF in physiological units of flow (mL/g/min) [81]. The ASL technique was originally developed to provide information about CBF, but recently several methods have been developed that enable the estimation of other physiological parameters such as the bolus arrival time and arterial CBV (Table 2) [82,83,84,85].

Several ASL sequences with different labeling schemes have been developed. The most widely used ASL sequences, continuous ASL (CASL) [86], pulsed ASL (PASL) [87] and pseudo-continuous ASL (pCASL), are shown in Figure 3B [88,89].

The CASL sequence uses a continuous Radiofrequency (RF) pulse in combination with a constant imaging gradient in the direction of the arterial blood flow to invert the arterial blood water spins. The main drawback of CASL is the requirement of long labeling RF pulses causing signal loss due to strong magnetization transfer (MT) effects and high specific absorption rates [80,81], historically limiting the applications to the acquisition of a single slice. To correct for MT effects in multiple slices, a separate labeling coil can be used (two-coil ASL) [90] or additional RF pulses during the control acquisition can be applied [91]. In rats, a dedicated neck labeling coil has been used to minimize MT effects and improve the SNR [86,92]. Due to their smaller size, two-coil CASL has proven to be more challenging in mice. When the dedicated coil is placed on the neck of a mouse, the short distance between the two coils will result in partial saturation of the brain signal. To overcome this limitation, an alternative approach was developed where the dedicated labeling coil is positioned at the heart region of the animal [93]. Even though CASL has a higher sensitivity than its pulsed counterpart, the use of CASL is limited due to the need for hardware providing continuous RF, a dedicated labeling coil and a high specific absorption rate.

In PASL, the labeling is achieved using a pulse or a pulse train of short RF inversion pulses. Depending on how the labeling is applied, the PASL sequences can be subdivided into two groups: the techniques that perform the labeling symmetrical with respect to the measured plane, called flow alternating inversion recovery (FAIR) [94], and the asymmetrical PASL sequences [87,95]. The PASL techniques have a higher inversion efficiency, a lower specific absorption rate and smaller MT effects than CASL. This, in combination with its ease of implementation and robustness, makes PASL the most widely used ASL sequence [96]. Due to the ease of implementation, FAIR PASL is the most frequently used PASL technique in pre-clinical studies.

More recently, the pCASL sequence was developed to combine the advantages of CASL and PASL [89]. It uses a train of short RF pulses that mimic the long continuous RF pulse from CASL. This way, it combines the high SNR of PASL with the high labeling efficiency of CASL without the need for specific hardware to generate a long continuous labeling pulse. pCASL is less dependent on flow velocity and shows better reproducibility than PASL and CASL [96]. Furthermore, it also shows better control of the MT effects and increased label efficiency compared to CASL [89]. The implementation of multi-slice pCASL is straightforward and does not require a dedicated labeling coil. Disadvantages are the lower label efficiency and higher specific absorption rate compared to PASL and CASL. Furthermore, pCASL has an increased sensitivity to magnetic field inhomogeneities in the labeling plane [97]. This becomes an increasing problem at the higher field strengths used in pre-clinical small-animal MRI. Even though pCASL is the recommended implementation of ASL for clinical applications [98], it is rarely used in pre-clinical research due to the lack of commercially available pCASL sequences on high field pre-clinical systems [99].

The major disadvantage of ASL in general is the low intrinsic SNR. Only a small part of the brain tissue volume consists of blood (2–4%), while the rest of the volume is filled with stationary tissue. Therefore, the signal of the labelled blood water spins represents only of a very small fraction of the overall water volume in the tissue. Additionally, the longitudinal T1 relaxation time of tagged blood water spins at clinical field strengths is similar to the arterial transit time [98]. This causes the labeling of the blood water spins to decrease by the time the measurement is performed resulting in lower SNR. The blood longitudinal relaxation time increases linearly with the magnetic field strength [100]. Therefore, an increase in the magnetic field results in a decreased labeling loss due to the decay of the blood water spins. On the one hand, the high magnetic field strength and the increased gradient strength on pre-clinical scanners improve the SNR and the resolution of the ASL images [101]. On the other hand, the higher magnetic field strength results in an increased T1 for gray matter, limiting SNR improvement [102]. The reduction in the T2* of blood, due to stronger susceptibility effects of paramagnetic deoxygenated hemoglobin at higher fields strengths, can lead to a decreased contribution of the capillaries to the ASL signal, which can lead to an underestimation of the CBF [103]. Furthermore, the increased magnetic field strength leads to increased contributions of magnetic field inhomogeneities in the labeling plane. The combination of these effects and several artifacts, such as subtraction errors, motion artifacts and susceptibility artifacts, make ASL a low SNR technique. To improve the SNR, several advanced pulse sequence or readout techniques, such as background suppression techniques [104,105], B0-correction [97,106,107], MT effect reduction [108] and motion correction techniques [109], have been proposed.

Nevertheless, ASL is increasingly used in pre-clinical research, in particular in models of neurological and neurodegenerative diseases [110,111,112,113]. In contrast to exogenous-perfusion-weighted imaging methods, ASL can be used to measure relatively rapid changes in CBF, for example, in response to changed physiological parameters such as pCO_2_ or pH. Exogenous-perfusion-weighted imaging methods would require first a clearance of the contrast agent from the blood pool. Such ASL experiments have been used to determine the cerebral vascular response to hypercapnia/hypoventilation (Figure 4), which was proposed as an earlier marker for the detection of vascular disfunction in neurodegenerative diseases [111].

Recently, a novel labeling scheme called velocity-selective ASL (vsASL) has been proposed [115]. In vsASL, the labeling is based on the velocity of the blood flow instead of the spatial location of the arterial spins. This method was specifically developed to address the cerebral blood flow in situations of slow flow, where the arterial transit time can be longer than the T1 relaxation time. However, to our knowledge, this method has not been implemented on a high field pre-clinical scanner.

### 3.4. Intravoxel Incoherent Motion

Intravoxel Incoherent Motion (IVIM) MRI, first described by Le Bihan et al. [116], is a diffusion MRI technique that can provide information about both molecular diffusion and perfusion at the same time [117]. The term IVIM stands for the motion of water within a voxel during an MR experiment. In biological tissue, this motion consists of contributions from the molecular diffusion of water and the microcirculation of blood in the microvascular network [116]. Originally, IVIM MRI was introduced based on the assumption that blood flow in randomly oriented capillaries mimics a random walk, similar to the Brownian motion of water molecules. This so-called pseudo-diffusion process, characterized by the pseudo diffusion coefficient (D*), results in an additional signal attenuation in diffusion-weighted imaging measurements, making diffusion MRI sensitive to both diffusion and perfusion. In the presence of diffusion gradients, both phenomena result in a decay of the signal intensity. The signal attenuation increases with the degree of diffusion weighting that is applied (b-value) [117]. However, during the typically applied short diffusion encoding times, the IVIM signal cannot be attributed solely to the randomized motion in the microvasculature network. The signal attenuation of the perfusion component will also have a contribution of phase dispersion caused by incoherent blood flow velocities within the microvascular network.

The pseudo-diffusion coefficient, D*, related to the perfusion in the microvascular network, is much higher than the molecular diffusion of water in the parenchymal tissue. Therefore, D* results in a faster decay of the signal attenuation. The faster signal decay, in combination with the small fraction capillary blood flow in the tissue, limits the contribution of the pseudo-diffusion to low b-values. Hence, the signal attenuation at high b-values is solely a consequence of diffusion effects in the extravascular-extracellular space. In order to capture both phenomena, the acquisition of an IVIM experiment consists of a conventional diffusion-weighted sequence, where a wide range of b-values are used. Compared to ‘conventional’ diffusion-weighted imaging, a number of measurements at low b-values are added to visualize the IVIM effect resulting from the pseudo-diffusion in the microcirculation (Figure 5).

In general, the signal attenuation can be expressed as:S(b) = S_0_ [(1 − f_IVIM_) × F_diff_ (b)+ f_IVIM_ × F_perf_ (b)],(1)
where S(b) represents the signal intensity at a certain b-value; S_0_ the signal intensity without diffusion sensitizing gradients (b = 0 s/mm^2^); F_diff_ (b) and F_perf_ (b) correspond to the diffusion and the perfusion component; f_IVIM_ the fraction of flowing blood in a voxel (%) and (1 − f_IVIM_) the extravascular-extracellular space where only diffusion effects are considered, respectively.

Fitting the overall signal attenuation at multiple b-values allows the determination of f_IVIM_ and D* (Table 2). The IVIM parameters f_IVIM_ and D* provide information about the contribution of perfusion to the MRI signal and can be related to classical perfusion parameters [117]. The original IVIM model describes both the perfusion and the diffusion component by a mono-exponential model resulting in the classical bi-exponential model for the signal attenuation. Over the past few years, several fitting approaches and different models for both the diffusion and the perfusion component have been explored.

The diffusion component can be approximated using a mono-exponential model. However, this is not correct in the case of biological tissue. This non-Gaussian behavior becomes visible at high b-values. Due to the small scale of the IVIM effects, this can result in significant errors in the parameter estimation. Examples of frequently used models that address this non-Gaussian behavior are the Kurtosis Model [118], which is known to fail at very high b-values [119], and the bi-exponential diffusion model containing a fast and a slow diffusion component [120].

Originally, two perfusion IVIM models were proposed by Le Bihan et al. [116]. The mono-exponential model assumes that the direction of the blood flow changes several times during the measurement. This only holds true when long diffusion times are used or under conditions of fast flow and short vessel segments [118,121]. At short diffusion times or under conditions of slow flow and long vessel segments, the blood will remain in a single segment during the measurement. Under these conditions, referred to as the ballistic limit, the signal attenuation can be more accurately fitted using a sinc model [116,117]. However, in reality, the situation lies somewhere in between these two models. Several other models have been proposed to fit the more realistic intermediate situation [122,123].

Multi-compartment models have been shown to fit signal attenuation for the perfusion component more accurately [124], leading to several multi-compartment models being proposed [121,125,126]. Recently, an in vivo rat brain study showed that the signal attenuation of the perfusion component can be more accurately fitted using a bi-exponential IVIM model that splits the perfusion component into a slow vascular pool, resulting from blood flow in the capillaries and a fast vascular pool, resulting from blood flow in the medium-sized vessel [121].

The fitting of the signal attention can be performed using several curve-fitting approaches. The most common methods are the one-step and the two-step approach. The one-step approach fits signal attenuation at all acquired b-values and simultaneously extracts the IVIM parameters using a nonlinear least-squares algorithm. This approach is easy to use, but is very sensitive to noise and local minima and often results in an inaccurate estimation of the parameters. The more robust two-step approach relies on the magnitude difference between the diffusion coefficient (D) and D*. Since the pseudo-diffusion only has a significant effect at low b-values, the signal attenuation at high b-values can be fitted to extract D (Figure 5). In the second step, the data from all b-values are used to get f_IVIM_ and D*. The main limitation of these two-step approaches is that a threshold separating the perfusion and the diffusion component has to be chosen. The optimal threshold can vary strongly between the organ measured and different pathologies. To overcome the need to select a cut-off threshold manually, an iterative algorithm to select the optimal threshold was proposed [127]. Recently, more complex fitting methods such as the Bayesian-fitting approach have gained interest [128].

The improvement of MRI hardware and software, improving the SNR and the acquisition time, has quickly expanded the field of IVIM MRI in recent years. One of the key features of IVIM is that it can provide quantitative information without the use of contrast agents and the need for identification of an AIF. The IVIM technique is a promising tool providing complementary information to the classical perfusion measurements (Table 1). In particular, the addition of IVIM to ASL measurements could provide additional diagnostic information without the administration of a contrast agent. The perfusion in several organs, such as the liver [129], kidney [130,131] and pancreas [132,133], has been successfully studied using IVIM MRI. Due to the small cerebral perfusion fraction, the implementation and the correlation with classical perfusion measurements in the brain has been proven to be more difficult [129,134,135].

Several factors influence the accuracy and reproducibility of IVIM. Apart from the different processing approaches, there is great diversity in the acquisition schemes used for IVIM MRI. In practice, the b-values and their distribution are often chosen heuristically. In general, an increase in the number of b-values results in a decreased error in the parameter estimation. However, the amount of b-values is often limited due to long acquisition times. Furthermore, it is often difficult to obtain accurate measurements at low b-values due to instabilities in the gradient amplifiers and MRI hardware limitations [117]. Currently, an increasing number of efforts are ongoing to optimize the number and distribution of b-values [136,137].

Applications of IVIM in rodents are limited. The higher SNR resulting from the ultra-high magnetic field used in small animal MRI systems improves the accuracy of the fit and the parameter estimation. This allows for the identification of small differences, which is crucial when measuring cerebral perfusion. However, the limitations mentioned above in combination with the wide variations in magnetic field strength used in rodents result in significant variance in the parameters across studies. There is therefore need for more uniform acquisition schemes and post-processing approaches. This can lead to increased reliability and reproducibility of the IVIM parameters.

### 3.5. Vascular Space Occupancy

Vascular Space Occupancy (VASO) is a relatively new endogenous MRI technique that allows assessment of changes in the CBV based on the intrinsic T1 differences of tissue and blood (Table 1) [138]. A schematic representation of the VASO principle is shown in Figure 6. First, the spins of both blood and tissue are inverted using a spatially non-slice-selective global inversion pulse. After the global inversion pulse, the longitudinal magnetization relaxes back to its equilibrium at the spin-specific T1 relaxation rate [139]. The difference in T1 relaxation between tissue and blood is expressed in a time difference for their magnetization curves to cross zero. The VASO technique exploits this difference using an optimal inversion time (TI) for blood, referred to as blood nulling. When an image is acquired at the TI, the only signal which is detectable arises from the extracellular tissue. Assuming a constant water volume per voxel, the signal arising from the extravascular tissue is proportional to (1-CBV) [139]. The optimal TI can be determined from the repetition time (TR) and the T1 of blood.

The VASO signal mainly arises from the differences in CBV. However, the contrast can be affected by several factors such as the CBF, the extravascular Blood Oxygenation Level Dependent (BOLD) effect, the contribution of cerebrospinal fluid, inflow effects and magnetization transfer effects. Some of these confounding contributions can be reduced by optimizing the acquisition parameters. The contribution of the BOLD effect can be minimized by using the shortest possible TE, while the TR should be chosen to be long enough to avoid strong CBF contributions. Inflow caused by non-steady-state inflowing blood spins can be minimized by the use of RF coils that cover the entire body or the application of magnetization-reset and crushing-gradient techniques [139,140]. Due to exchange of water between blood and tissue, the inverted blood spins will affect the signal from the extravascular tissue. This will result in a lower SNR. This signal contrast between blood and tissue can be enhanced by the application of a magnetization transfer pulse [141].

The global inversion pulse used in VASO results in a low SNR. In general, the SNR is improved when higher magnetic field strengths are used. However, in VASO, there are several confounding factors that counteract this improvement. First of all, at higher magnetic field strengths, the difference between the T1 of tissue and blood will become smaller, reducing the sensitivity. Secondly, the higher field strength results in increased extravascular BOLD effects, counteracting the negative VASO signal and further reducing the sensitivity. This could underestimate CBV values. Furthermore, the efficiency of the blood nulling is limited due to increased field inhomogeneities, and the acquisition will become more prone to geometrical distortion and inflow artifacts [139,142]. To overcome these limitations, an alternative method using a slab-selective inversion instead of a global inversion was proposed [143]. This was further extended to slice-saturation slab-inversion VASO by applying additional RF pulses before the slab-selective inversion, which further improved the CNR compared to the original VASO method [142].

The main limitation of VASO is that it does not provide an absolute measurement of CBV. Therefore, VASO has been mainly used to measure CBV changes in functional MRI studies. In order to provide quantitative information about the CBV values, an alternative approach based on the T1 shortening effect of T1 contrast agents was developed (Table 1) [144]. Contrast-based VASO is a steady-state technique that relies on the assumption that the contrast agent remains confined to the intravascular space. Therefore, the T1 shortening effect is limited to the blood compartment. The acquisition consists of identical pre-and post-contrast experiments. Due to the T1 shortening, the blood signal from the post-contrast measurement will no longer be nulled at TI. Since the contrast agent remains intravascular, the T1 weighting of the tissue will not be affected in between the two experiments. The difference between the pre-and post-contrast images can be used to calculate the absolute CBV (Table 2).

Compared to DSC MRI, VASO MRI has several advantages and limitations. Since it is a steady-state technique, contrast-enhanced VASO does not require extremely short bolus injections and acquisition times. This allows a higher spatial resolution compared to a dynamic imaging method. Another advantage is that lower SNR of contrast-based VASO can be improved by signal averaging over different acquisitions, which is not possible in DSC [144]. Finally, contrast-enhanced VASO is less prone to geometric distortions and susceptibility artifacts and does not require the estimation of the AIF. The main disadvantage of contrast-based VASO is that it only provides information about the CBV, while DSC also provides information on the CBF and MTT (Table 2).

Recently, a new approach called inflow-VASO (iVASO) was introduced [145]. In iVASO, only the water spins flowing into the slice are nulled using spatially slice-selective inversion. The TI for the nulling of the blood water spins is similar to the transit time of arterial blood water spins to reach the capillaries. Therefore, the iVASO technique is no longer sensitive to the total CBV, but is mainly sensitive to the contribution of the arterial and arteriolar compartment. Some of the main limitations of the original VASO technique can be overcome by the modification of the VASO sequence. By using a spatially slice-selective inversion pulse, the tissue signal is not affected by the blood nulling, which significantly increases the SNR. The iVASO technique does not depend on the difference between the T1 of the tissue, making it more interesting for use at higher magnetic fields strengths. Furthermore, since no inversion is applied in the imaging slice, iVASO is less dependent on T1 changes in the tissue. This makes iVASO a more suitable technique to study CBV in pathological conditions. Finally, iVASO significantly reduces the contribution of cerebrospinal fluid.

The iVASO signal changes are highly dependent on the choice of the TI and TR. When short TR is used, iVASO will become mainly sensitive to arteriolar blood water spin effects. At longer TR, capillary blood water spin effects will play a role. Due to its dependence on the TI, iVASO requires prior knowledge about the arterial transit time to provide absolute measurements of the arterial CBV [145,146]. To overcome this limitation, a quantitative iVASO method was introduced [147,148]. This approach uses the consecutive acquisition of a control scan without blood nulling and an iVASO acquisition where the inflowing blood water spins are nulled. Since the tissue magnetization is the same in both acquisitions, the subtraction of these two images provides an absolute measurement of the arterial CBV (Table 2). The pulse sequence and the subtraction procedure are similar to ASL. However, the fundamental difference is that in iVASO, the water spins are nulled, whereas they are tagged in ASL [146].

The quantitative iVASO methods allow the completely non-invasive absolute measurements of CBV with a higher SNR than the originally proposed VASO technique (Table 1). Quantitative iVASO is a promising tool to study both normal tissue and tissue with altered perfusion. It provides an interesting addition or alternative to the conventional methods for measuring CBV (Table 2). To date, VASO MRI was mainly applied in clinical research. However, due to the advantages of the more recently developed iVASO, it may become a useful tool in future pre-clinical research.

### 3.6. Steady-State Susceptibility Contrast MRI

Steady-state contrast enhanced MRI (SSC MRI) is an exogenous MRI technique that uses a contrast agent with a long (blood) half-life in order to maintain a more stable contrast agent concentration during the experiment (Table 1). Most SSC MRI studies use ultra-small super-paramagnetic iron oxide nanoparticles (USPIOs) as blood pool contrast agents. USPIOs have a longer (blood) half-life than the frequently used gadolinium chelates, allowing measurements closer to steady-state conditions. Furthermore, they have a larger hydrodynamic diameter than gadolinium chelates and are therefore more likely to remain confined to the intravascular space [149].

In SSC MRI, the susceptibility-induced contrast difference between the intravascular and extravascular compartments can be used to assess the CBV. First, the transverse relaxation rate R2 (SE) or R2* (GE) is measured before the injection of a contrast agent and under the steady-state conditions after the administration of a contrast agent is measured. Then, the linear relation between the changes in the transverse relaxation rate and the intravascular contrast agent concentration are used to generate an index, the relative CBV (rCBV).

Unlike DSC MRI, there is no need for rapid acquisition during the first pass of the bolus. Therefore, in the steady-state approach, a higher SNR and thus a higher spatial resolution can be achieved. The long half-life of the contrast agent allows for dynamic CBV measurements in functional MRI studies [149,150]. The superparamagnetic properties of the USPIOs result in significant enhancement transverse relaxation rates compared to gadolinium base contrast agents. The larger diameter of iron oxide particles might result in slower leakage out of the vessel, even in case of BBB disruptions, reducing the error in the rCBV measurements [151,152].

A limitation of SSC MRI is that no information about the CBF and MTT can be obtained (Table 2). SSC MRI assumes that the susceptibility-induced contrast difference is independent of its neighboring voxels. However, this could lead to an overestimation of rCBV in the vicinity of large vessels. The steady-state approach has been commonly used in pre-clinical studies to measure rCBV. The major limiting factor for translation to the clinic is the lack of clinically approved contrast agents with a long enough half-life for steady-state measurements.

### 3.7. Vessel Size Imaging

Vessel size imaging is an exogenous MRI technique that provides a measurement of the average microvessel size within a voxel by comparing changes in the ΔR_2_ (GE) and ΔR_2_* (SE) relaxation rates induced by the injection of a blood pool contrast agent. The quantitative information about the average vessel diameter within a voxel is represented by the Vessel Size Index (VSI) (Table 2). Vessel size imaging can provide valuable information about the microvascular density that was previously only available from non-invasive measurements such as biopsies [153].

Vessel size imaging exploits the difference in response of the ΔR2 and ΔR2* transversal relaxation rate, as the vessel diameter increases [154]. The dimensionless ratio ΔR2*/ΔR2, which increases with the vessel diameter, can be used to provide information on the average vessel size within a voxel [76]. However, this dimensionless ratio depends on the concentration of the contrast agent. To avoid this dependence, the mean vessel density (Q), defined as the ratio ΔR2/(ΔR2*)^2/3^, was introduced. Under conditions of appropriately high contrast agent concentrations and long echo time, Q has shown a good correlation with histological measurements of the vessel density [155]. Expanding on this, the VSI (µm) was introduced to provide a quantitative measurement of the average vessel size within a voxel. In order to measure the VSI from the ΔR_2_* and ΔR_2_ quantitatively, other parameters such as the water diffusion rate, D, the contrast agent concentration and the absolute blood volume fraction (CBV) should be considered.

Vessel size imaging has two types of applications, steady-state vessel size imaging [156] and dynamic vessel size imaging [157]. Similar to SSC MRI, the steady-state method uses a contrast agent with a long blood half-life. The absolute CBV cannot be measured in the steady-state approach. However, under the assumption that the contrast agent remains intravascular during the measurement, and the concentration of the contrast agent in the blood is known, the blood volume fraction can be determined from ΔR2*. When the contrast agent concentration, which is proportional to the susceptibility difference between blood and brain tissue Δχ, and the diffusion coefficient D are determined, the absolute VSI can be determined using the relation (ΔR2/ΔR2*)^3/2^ (D/Δχ)^1/2^. The steady-state approach involves an invasive measurement of the blood contrast agent concentration, eliminating the need to measure absolute CBV. The value of D can be determined by performing a separate diffusion measurement or chosen based on prior knowledge. An advantage of the steady-state approach is the higher achievable spatial resolution using longer acquisition times. Similar to SSC MRI, the steady-state approach is mainly used for pre-clinical research, due to the lack of clinically approved contrast agents with a long enough blood half-life.

The dynamic approach is an extension of DSC MRI that acquires both GE and SE measurements during the first pass of the bolus of a contrast agent. In the dynamic approach it is impossible to measure the concentration of the contrast agent in the blood. However, in contrast to the steady-state approach, the absolute blood volume fraction can be determined from the susceptibility difference between blood and brain tissue Δχ obtained from ΔR2* measurements, as in conventional DSC, eliminating the need to measure the concentration of the contrast agent in the blood. When the absolute CBV and the diffusion coefficient D are determined, the quantitative VSI can be obtained using ΔR2/(ΔR2*)^3/2^ (CB D)^1/2^.

The dynamic approach requires the acquisition of both GE and SE measurements during the first pass of the contrast agent bolus. This limits the spatial resolution that can be achieved. In order to improve the spatial resolution, a method using separate GE and SE acquisitions and a dual injection of contrast agent was proposed [158]. However, this technique has several other disadvantages such as the need of a higher dose of contrast agent, longer acquisition times and the potential influence of the first injection of the contrast agent on the second measurement.

To date, vessel size imaging has mainly been used in tumor research. However, vessel size imaging can also be explored as a promising tool in other neurodegenerative and neurological diseases where changes in the vessel size or density are expected. The VSI provides additional information on the mechanism behind perfusion changes in the brain complementing other perfusion measurements.

### 3.8. SAGE-Based DSC MRI

Recently, a combined SAGE Echo-Planer Imaging (SAGE EPI) MRI pulse sequence that allows the simultaneous measurement of spin-echo and gradient-echo DSC MRI without the need for sequential acquisition of two experiments was developed [79,157,159]. Using a multi-echo approach, this technique provides measures of the typical DSC MRI parameters (CBF, CBV, MTT), the kinetic permeability parameters (K^trans^, ν_e_) and the relative vessel size (VSI) (Table 2). Unlike conventional DSC measurements, the SAGE-based approach is independent of T1 effects [79].

The SAGE-based DSC MRI technique combines the advantages of several methods into a single acquisition. It provides complementary perfusion measures obtained from first pass measurements using a clinically approved contrast agent, making SAGE-based DSC MRI a promising tool to study neurodegenerative diseases (Table 1). However, SAGE-based DSC MRI also has its limitations.

Some of the limitations, such as the need to determine an AIF and geometric distortions, the dependence on the injection protocol and the need for extremely short acquisition times, have already been mentioned in the description of the conventional perfusion techniques. In addition to these limitations, the use of DSC MRI has been limited by the need for short TEs to measure changes in R2 and R2* accurately [160]. In particular, the first TE used in SAGE-based MRI should be short enough, since the T2 and T2* become very short during the bolus of the contrast agent. Parallel imaging methods can be used to shorten the TE. However, on high-field pre-clinical small-animal systems, often a limited number of channels is available, limiting the acceleration that can be obtained using parallel imaging. Therefore, alternative approaches using partial Fourier encoding have been proposed [77]. The implementation of other acceleration methods such as the keyhole method [161] and multiband or simultaneous multi-slice excitation [162] is highly beneficial for future pre-clinical studies, especially since several acceleration methods can be combined with each other and with parallel imaging, further reducing the scan time and the TE [160].

### 3.9. Phase Contrast Flow MRI

Phase Contrast MRI (PC MRI), also referred to as velocity mapping, is an endogenous MRI technique that uses flow-encoding gradients to visualize and quantify the velocity of moving fluids [163,164]. It relies on the dephasing of moving spins when they are subjected to a bipolar gradient. The net phase shift of the moving spins will be proportional to the velocity of the spins along the direction of the bipolar gradient. Spins in the same direction of the bipolar gradient will obtain a positive net phase shift, while spins moving in the opposite direction will acquire a negative net phase shift [164]. In order to compensate for unwanted phase shifts induced by other sequence parameters and to remove the background signal, the phase images are subtracted by a flow compensated reference image that was acquired with the same acquisition parameters but with an inverted bipolar gradient [164,165]. The PC MRI technique is only sensitive to flow along the direction of the bipolar gradient. In order to obtain measurements of flow in arbitrary directions, flow encoding gradients can be applied along all three orthogonal directions. However, the addition of measurements along multiple axes significantly increases the acquisition time for PC MRI measurements.

Since the velocity-encoded images contain flow measured both in the direction off and in the opposite direction of the bipolar gradient, a phase range spanning from -π to π is chosen. Velocities corresponding to shifts larger than |π| will induce phase wrapping or aliasing in the velocity encoding image. Therefore, the maximum velocity that can be measured in a PC MRI experiment is limited. The maximum velocity that can be measured without the occurrence of aliasing at a certain gradient strength is expressed by the velocity encoding parameter (VENC). Before a PC MRI experiment, the VENC, which is inversely proportional to the gradient strength, should be chosen so that the maximum velocity corresponds to a phase shift of 180° to avoid aliasing. The correct choice of the VENC parameter, which should be estimated based on the velocities of interest, is crucial for the accuracy of PC MRI measurements. When the VENC is too high, the flow in the velocity-encoded image will be compressed in a small range of phase shifts. The inability to distinguish between small velocity differences will decrease the SNR and the accuracy of the images [165]. This might lead to inaccurate measurements, especially in regions with slow flow [166]. A too-low VENC will result in aliasing, making quantitative measurements difficult. The VENC is inversely proportional to the strength of the bipolar gradient and can be adapted by changing the bipolar gradient strength [163]. However, a decrease in the field strength will be accompanied by a decreased SNR.

The main applications of PC MRI in the brain are flow measurements of cerebral spinal fluid [164,167] and vascular imaging using phase contrast MR angiography (PC MRA) [168]. When introduced, the PC MRA technique was mainly used to visualize large vessels with active blood flow non-invasively [169,170]. This can be achieved with 3D cine PC MRA or 4D PC MRA, which is discussed in detail elsewhere [170,171].

Rather than just generating angiograms, PC MRA can be used to measure the velocity and the direction of blood flow within the vascular network. In order to quantify the total global CBF in the brain non-invasively, the major cerebral feeding vessels 2D PC MRA are used [172,173,174,175]. This is done by placing a thin imaging slice perpendicular to the main feeding arteries of the brain. In 2D PC MRA, the total CBF can be measured using either cardiac-gated or non-cardiac-gated acquisition, which significantly reduces the acquisition time [176,177,178]. The total CBF, containing the entire blood supply to the brain, obtained with PC MRA can be used to normalize other CBF mapping techniques such as DSC [179] and ASL [180] to provide absolute CBF maps. This diminishes some confounding factors, such as the need to identify an AIF or the labeling efficiency, typically hindering the absolute CBF quantification [175,181]. When used in combination with measurements for quantification of blood oxygenation, it can be used to estimate the cerebral metabolic rate of oxygen (CMRO_2_) [182,183]. The CMRO_2_ is a key measure of cerebral functioning. Alterations in CMRO_2_ have been suggested to play an important role in several neurological disorders (19). In contrast to Positron Emission Tomography (PET), which is considered as the golden standard for CMRO_2_ quantification, PC MRI provides a completely non-invasive way to measure CMRO_2_.

Several limiting factors should be considered when 2D PC MRA is used to measure flow velocities. In order to describe the velocity accurately, the acquisition slice should be positioned perpendicular to the vessel orientation [163]. When the slice is not positioned perpendicularly to the vessel, saturation effects might occur due to in-plane flow. Furthermore, the acquisition slice should be placed in a straight vessel segment with laminar flow to avoid intravoxel dephasing. Another limiting factor is the need to determine the optimal VENC, which requires prior knowledge or a good approximation of the maximum velocities in the vessel of interest.

### 3.10. Quantitative Susceptibility Mapping

Susceptibility Weighted Imaging (SWI) exploits the magnetic susceptibility differences between tissues, which are often the cause of artifacts in MR images, to obtain images with increased contrast [184]. It exploits the complementary information about the structure and function of tissues contained in phase images by combining magnitude and phase data (Figure 7). The acquisition usually consists of a flow compensated T2*- weighted 3D gradient echo sequence. SWI is commonly used for tissue characterization based on susceptibility differences caused by deoxygenated blood, iron deposition and calcification in neurological disorders [184,185]. The SWI approach has several limitations. It is highly dependent on imaging parameters, suffers from blooming artifacts and has the orientation dependence of the phase signal. This limits SWI to mainly qualitative measurements [186]. To overcome these limitations, Quantitative Susceptibility Mapping (QSM) was developed (Table 1). QSM is a post-processing technique that quantifies the underlying magnetic susceptibilities based on the phase images.

The computation of the magnetic susceptibility consists of a three-step process: First, the magnetic field is estimated from the raw phase data using a phase unwrapping algorithm [187,188]. Next, in order to get a map solely consisting of the susceptibility sources inside the brain or region of interest, the contributions from outside the field of view are removed by background field removal [189,190,191]. Finally, the inverse problem from field perturbation to magnetic susceptibility has to be solved.

The magnetic field in perturbation in each voxel, caused by the magnetization of tissue when placed into an external magnetic field, can be approximated as a magnetic dipole producing a dipole field that extends beyond the voxel itself. Therefore, the magnetic field perturbation in a certain voxel is a superposition of its own dipole field and that of its neighboring voxels [192]. When the susceptibility distribution is known, the field perturbation can be obtained by the convolution of the susceptibility distribution and the field of a unit dipole (i.e., dipole kernel) [193]. However, multiple susceptibility distribution could result in the same field perturbation, making this is an ill-posed inversion problem. A simple kernel division would cause errors that would be represented as streaking artifacts in the reconstructed susceptibility map [194,195]. Several methods to overcome this ill-posed inversion problem have been proposed.

The ill-posed problem can be solved analytically by repeated acquisition of the magnetic field perturbation at different orientations relative to the magnetic field. This method is referred to as calculation of susceptibility through multiple orientation sampling, or COSMOS [196], and is regarded as the golden standard for QSM. This requires, however, long acquisition times and the assumption of isotropic magnetic susceptibility, thereby making this technique impractical for in vivo measurements [193,197]. Therefore, regularization approaches using prior information to determine a unique solution from a single acquisition are used. Several regularization algorithms for solving the ill-posed inversion problem from a single acquisition and minimizing the streaking artifacts have been proposed [194,198,199,200,201].

The QSM post-processing technique has been used to provide quantitative measurements of the cerebral mixed venous oxygenation saturation (SvO_2_) (Table 2). The SvO_2_ can be calculated by exploiting the magnetic properties of hemoglobin. When fully oxygenated, diamagnetic hemoglobin results in negative susceptibility, leading to a large susceptibility decrease in arteries. In contrast, deoxygenated hemoglobin is paramagnetic. Therefore, the venous vessels will result in increased susceptibility [202]. The ratio of blood oxygen defined as the Oxygen Extraction Fraction (OEF) can be obtained by calculating the difference between the susceptibility in the veins and tissue (Table 2). Additionally, when used in combination with other MRI sequences that provides a measure of the CBF, QSM can provide a completely non-invasive quantitative measurement of CMRO_2_ (Table 2) [197]. QSM can provide important complementary information to other perfusion MRI techniques and has high potential to be used as an early biomarker for small changes in the cerebral physiology [197].

A major limitation of QSM is the assumption of the isotropic magnetic susceptibility within a voxel. However, this assumption does not hold for all types of brain tissue. In order to account for tissue containing anisotropic magnetic susceptibilities, tensor imaging was developed [203,204]. Another important limitation of QSM is that it can only provide a relative quantification of the magnetic susceptibility [205,206]. So far, this has limited the inter-subject comparison of QSM measurements. The accuracy of QSM is strongly dependent on several factors, which requires careful standardization and optimization of the post-processing and acquisition parameters to increase its reproducibility and accuracy further [205]. To date, QSM has mainly been used in the human brain, but it might also become useful in other parts of the body.

Only a few studies have investigated QSM in rodents [207,208,209]. Pre-clinical QSM could provide interesting insight in brain functioning under altered physiological conditions. Comparing data under normoxic, hyperoxic and hypercapnic conditions, and verification with blood gas analysis, could improve our understanding of neurological disease mechanisms and cerebral functioning. QSM especially benefits from a high magnetic field strength, as higher field strength increases both the SNR and the contrast in the phase images.

### 3.11. Quantitative Blood-Oxygenation-Level-Depedent MRI

The BOLD MRI technique is based on the different magnetic properties of oxygenated and deoxygenated blood. During brain activation, the neurons consume an increased amount of oxygen. This results in a local increase in the regional CBF towards the site of activation. When placed in an external magnetic field, the deoxyhemoglobin in the blood vessels will induce a magnetic susceptibility difference between blood and tissue. This susceptibility difference results into signal changes of the local contrast in both T2- and T2*-weighted images and can therefore be used as an endogenous contrast agent to measure brain activation indirectly [210]. The temporal resolution of BOLD MRI is too poor to perform direct measurements of neuronal activity in the brain. However, due to its high spatial resolution and the ability to measure the much slower response in regional blood flow, BOLD fMRI has been used extensively as a tool to investigate temporal MR signal changes non-invasively during brain activity as a response to a cognitive tasks or stimuli under both normal and pathophysiologic conditions [211,212].

However, the BOLD effect can also be used to study the resting state or baseline of the brain and its impairment by neurodegenerative diseases. This allows quantitative evaluation of cerebral blood oxygenation and was therefore named quantitative BOLD (qBOLD) (Table 1) [213]. The deoxyhemoglobin present in the blood vessels creates tissue specific mesoscopic field inhomogeneities [214]. Using a Gradient Echo Sampling of Spin Echo (GESSE) sequence, the mesoscopic field inhomogeneities can be separated from microscopic and macroscopic inhomogeneities [215]. The mesoscopic, tissue specific BOLD signal can then be fitted to a multi-compartment model based on prior knowledge about the brain tissue. This allows derivation of quantitative hemodynamic parameters, such as the deoxygenated blood volume on the venous site of the blood vessel network (DBV) and the brain tissue OEF [213] (Table 2). The results from qBOLD are in good correspondence with direct measurements of blood oxygenation in a validation study in a rat model [216].

The original qBOLD method has some limitations. An additional field map needs to be acquired to correct for signal loss near air-tissue interfaces. This becomes increasingly important when imaging rodents due to the smaller brain size, the close location of sinuses to the brain and the associated decreased shim quality high magnetic field strength. Furthermore, the use of multi-compartment models, removing the R2-weighting and nulling the cerebrospinal fluid signal, requires a high SNR due to the large amount of modeling parameters. An alternative method using a FLAIR-GASE method was recently proposed [217]. However, this method has not been validated in animal models yet. Furthermore, a dynamic approach using a parallel acceleration technique to provide a higher temporal resolution compared to the GESSE sequence, named multi-echo asymmetric spin echo (MASE), was proposed [218]. This method can be used to measure dynamic changes in the brain oxygenation during brain activation.

Similar to QSM, the combination of qBOLD with another technique that provides measurements of the CBF allows quantitative measurement of CMRO_2_. The estimation of CMRO_2_ in QSM and qBOLD relies on a number of assumptions. Recently, a combined QSM+qBOLD (QQ) approach was proposed [219]. The QQ approach allows the quantification of OEF and CMRO_2_ without the need for altered physiological conditions or empirical assumptions. This combined approach showed clearer gray and white matter compared to separate acquisitions of QSM and qBOLD, and a more uniform OEF and better agreement with independent methods to estimate CMRO_2_, compared to QSM. This makes QQ a highly promising method for improvement in the accuracy of blood oxygenation measurements in the brain.

## 4. Conclusions

Perfusion MRI is a highly versatile non-invasive imaging technique, widely used in clinical and pre-clinical investigation of the cerebral microcirculation. Pre-clinical perfusion MRI has the potential to overcome some of the shortcomings of clinical perfusion studies. Pre-clinical MRI is an important translational tool for the development and validation of novel MRI sequences to study changes in the cerebral microvasculature. Future advancements facilitated by the novel perfusion methods, advanced techniques such as parallel imaging, acceleration methods, artificial intelligence and in particular increased standardization of perfusion MRI protocols will result in increased reliability and reproducibility of perfusion parameters in high-field small-animal MRI. This will increase the translational value of pre-clinical perfusion MRI and may result in a better insight in pathophysiological disease mechanisms associated with changes in the cerebral microcirculation.

## Figures and Tables

**Figure 1 diagnostics-11-00926-f001:**
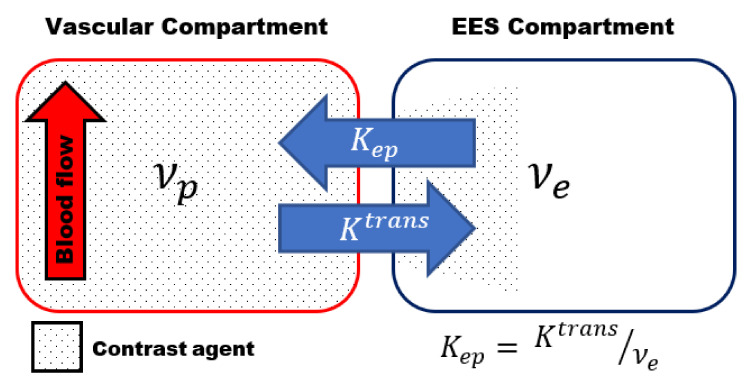
Schematic representation of the two-compartmental pharmacokinetic (PK) Tofts model comprising an intra-vascular compartment (left) and extravascular -extracellular space (EES) compartment (right). The volume transfer between the two compartments is depicted by the transfer constant between the blood plasma and the EES (K^trans^) and the transfer constant between the EES and blood plasma (k_ep_), respectively. k_ep_ can be calculated using k_ep_ = K^trans^ /ν_e_, whith ν_e_ volume of the EES per unit volume of tissue.

**Figure 2 diagnostics-11-00926-f002:**
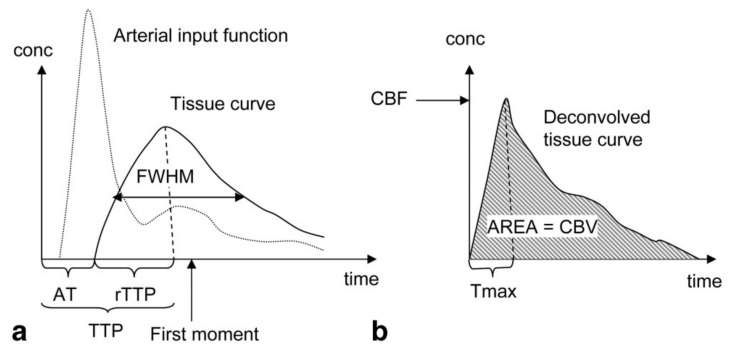
Parameters derivation in Dynamic Susceptibility Contrast Enhanced (DSC) MRI: (**a**) Semi-quantitative interpretation can be derived from the signal intensity curve. The Arrival Time (AT) of the bolus can be determined from the time interval between the injection of the contrast agent and the time point where the contrast agent is first detected in the tissue. The Time-To-Peak (TTP) is determined by the time interval between the injection of the contrast agent and the peak of the contrast agent in the tissue. The Full Width at Half Maximum (FWHM) is dependent on the Mean Transit Time (MTT) of the tissue. (**b**) the contrast concentration curve can be calculated by deconvolution of the Arterial Input Function (AIF) and the signal intensity curve. The Cerebral Blood Flow (CBF) is determined by the maximum height of the contrast concentration curve (T_max_). The Cerebral Blood Volume (CBV) is determined by the area under the contrast concentration curve, while the MTT can be calculated using MTT = CBV/CBF. Reproduced with permission from Leif Østergaard, Journal of Magnetic Resonance Imaging, published by John Wiley and Sons, 2005 [68].

**Figure 3 diagnostics-11-00926-f003:**
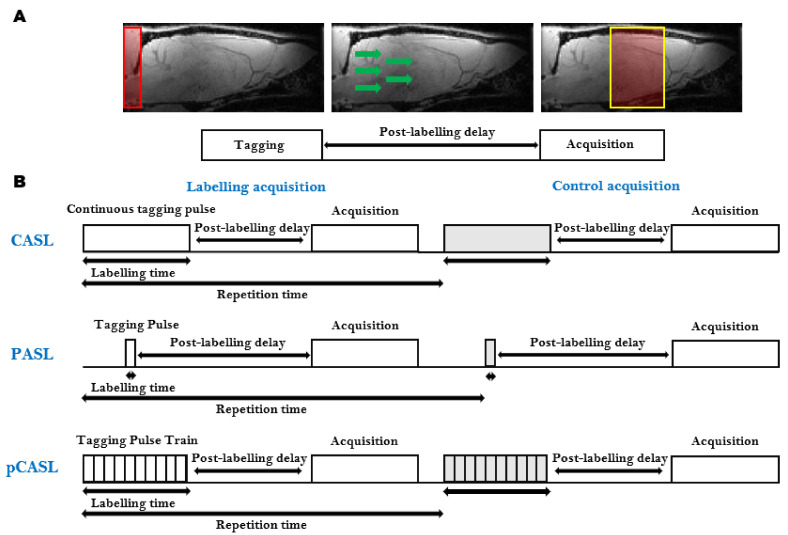
Arterial Spin Labeling (ASL): (**A**) The general ASL scheme consists of two consecutive acquisitions. The first acquisition consists of the tagging of the arterial blood spins (Left), followed by a post-labeling delay time, during which the tagged spins flow to the tissue of interest and exchange with the stationary spins in the extravascular-extracellular space (Middle), and finally, the image is acquired (Right). The second acquisition, which serves as a control, is identical to the first acquisition without the tagging of the arterial water spins; (**B**) Schematic representation of the most widely used ASL sequences. Continuous ASL (CASL), uses a continuous Radiofrequency (RF) pulse in combination with a constant imaging gradient in the direction of the arterial blood flow to invert the water spins of arterial blood (Top). In Pulsed ASL (PASL), the labeling is achieved using a pulse or a pulse train of short RF inversion pulses (Middle). pseudo-Continuous ASL (pCASL) uses a train of short RF pulses that mimic the long continuous RF pulse from CASL (Top).

**Figure 4 diagnostics-11-00926-f004:**
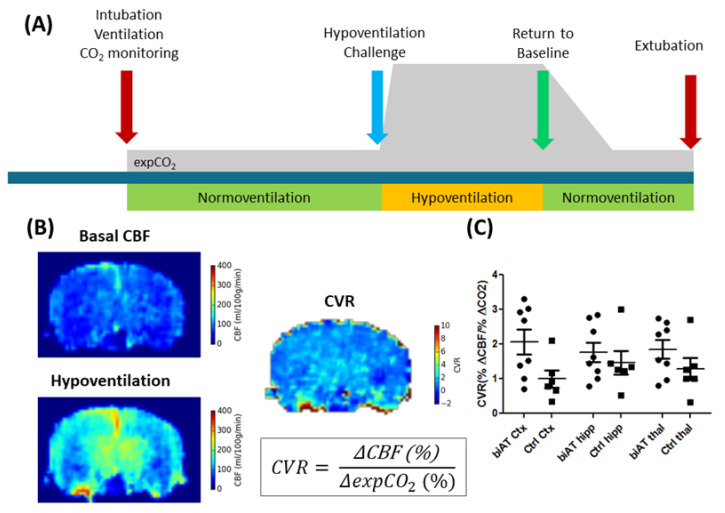
ALS experiments in control mice (Ctrl) and an Alzheimer disease model (biAT) according to [111]. (**A**) Illustration of the experimental paradigm. Animals were intubated, and ventilation was controlled. ALS measurements commenced during a period of normoventilation. In order to assess the cerebral vascular response (CVR) to a hypercapnic challenge, animals were hypoventilated. This was followed by a control measurement under normoventilation. Expired CO_2_ levels (expCO2) were recorded during the experiments. More details on the protocol can be found in [111,114]. (**B**) CBF maps and calculation of the CVR based on the baseline CBF and the CBF acquired under hypoventilation. (**C**) Comparison of CVR in the cortex (Ctx), hippocampus (hipp) and thalamus (thal) of control mice and biAT mice. This figure is based on data presented in [111].

**Figure 5 diagnostics-11-00926-f005:**
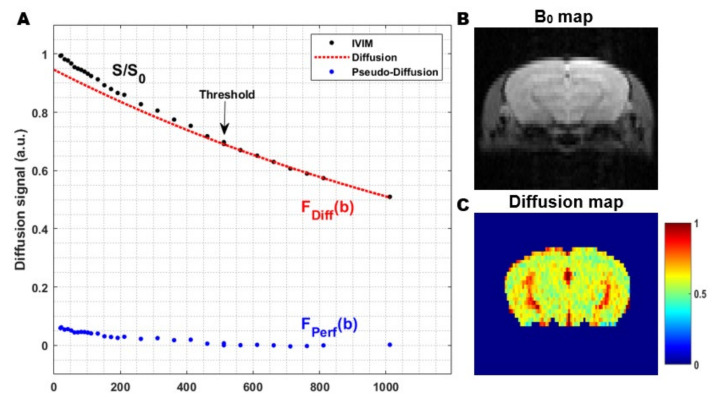
The general principle of the two-step fitting approach in Intravoxel Incoherent Motion (IVIM) MRI: (**A**) In a first step, the signal attenuation at high b-values is fitted (red). Since the signal attenuation at high b-values only contains contributions from the molecular diffusion of water in the extravascular-extracellular space (EES), this can be used to estimate the diffusion coefficient. In the second step, the data from all b-values are used to extract the other IVIM parameters. The contribution of the pseudo-diffusion in the microvascular network (blue) can be visualized by subtracting the diffusion contribution from the total IVIM signal (black-red); (**B**) Coronal B_0_ map in a rat and; (**C**) Corresponding diffusion coefficient map, obtained from the first step in the two-step fitting approach. The images were acquired from a mouse brain using a single shot gradient spin-echo Echo Planar Imaging (EPI) sequence at a magnetic field strength of 9.4 T (Echo Time 37 ms, Repetition Time 1000 ms, flip angle 90°, 3 orthogonal diffusion gradient directions, gradient separation (Δ) 1.6 ms, gradient duration (δ) 7.9 ms and 30 b-values 0–1000 s/mm^2^).

**Figure 6 diagnostics-11-00926-f006:**
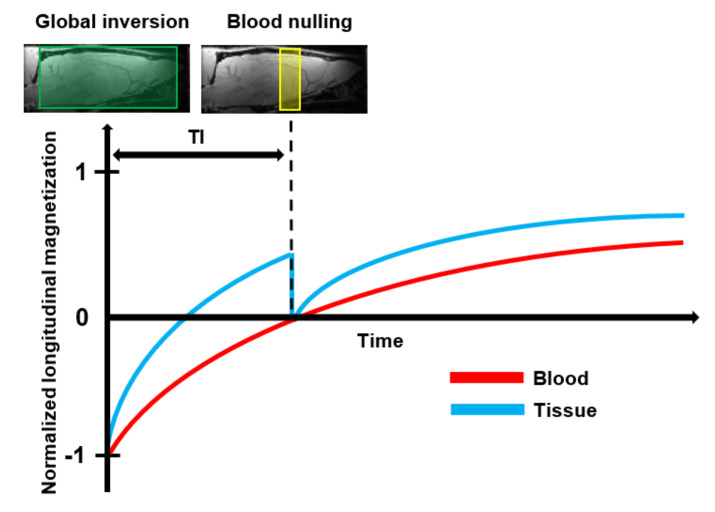
Schematic representation of the basic principle of Vascular Space Occupancy (VASO) MRI: First, the spins of both blood and tissue are inverted by the use of a spatially non-slice-selective global 180° inversion pulse. After the global inversion pulse, the longitudinal magnetization of blood and tissue relax back at their spin-specific T1 relaxation rate. At the optimal Inversion Time (TI), when the longitudinal magnetization of the tissue crosses zero, an additional 90° excitation pulse is applied to null the blood signal. Then, an image is acquired that contains only signals that arise from the extravascular tissue.

**Figure 7 diagnostics-11-00926-f007:**
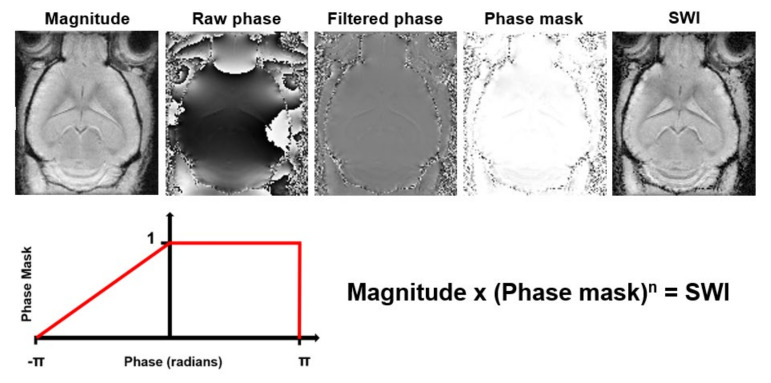
Processing steps to generate Susceptibility Weighted Images (SWI) from magnitude and phase data. The raw phase data are filtered to remove low frequency fluctuations. Then, a phase mask that scales the filtered phase images to a range from 0 to 1 is created. This phase mask is then multiplied several times (n) with the magnitude image to generate the SWI with enhanced contrast.

**Table 1 diagnostics-11-00926-t001:** Overview of the MRI techniques discussed in this article.

Exogenous	Endogenous
DCE DSC Contrast-based VASO SSC MRI Vessel size imaging SAGE-based DSC	ASL IVIM VASO iVASO PC MRI QSM qBOLD

Abbreviations: DCE: Dynamic Contrast Enhanced; DSC: Dynamic Susceptibility Contrast Enhanced; SSC: Steady-State Susceptibility Contrast; SAGE: Spin- And Gradient-Echo Echo-Planer; ASL: Arterial Spin Labeling; IVIM: Intravoxel Incoherent Motion; (i)VASO: (Inflow) Vascular Space Occupancy; PC MRI: Phase Contrast MRI; QSM: Quantitative Susceptibility Mapping; qBOLD: quantitative Blood-Oxygenation-Level-Dependent.

**Table 2 diagnostics-11-00926-t002:** Summary of the different output parameters obtained by the different MRI techniques discussed in this article.

Technique	Frequently Used Sequence	Output Parameters	Contrast Agent	Main Disadvantages
DCE	T1-weighted GE EPI	IAUGC K^trans^ ν_e_ k_ep_	exogenous	AIF K^trans^ combined measure
DSC	T2-weighted SE or T2*-weighted GE EPI	CBF CBV MTT (K^trans^, ν_e_, k_ep_)	exogenous	AIF first pass measurement
ASL	2D EPI or 3D GRASE	CBF Bolus arrival time Arterial CBV	endogenous	low SNR
IVIM	PGSE EPI	D D* f_IVIM_	endogenous	long acquisition time different approaches
VASO	T1-weighted GE, SE or GRASE EPI	CBV	exogenous/endogenous	low SNR only measurement of CBV
iVASO		arterial CBV	exogenous	
SSC	T2 -weighted SE or T2*-weighted GE EPI	relative CBV	exogenous	only measurement of CBV lack of clinically approved contrast agents
Vessels size imaging	SAGE EPI	VSI	exogenous	first pass measurement or lack of clinically approved contrast agents
SAGE-based DSC	SAGE EPI	CBF, CBV, MTT K^trans,^ ν_e_, k_ep_ VSI	exogenous	AIF first pass measurement
2D PC MRA	T2-weighted GE	total CBF (CMRO_2_)	endogenous	only measurement of global CBF choice of optimal VENC
QSM	3D multi-echo GE	OEF (CMRO_2_)	endogenous	assumes isotropic magnetic susceptibilities relative quantification
qBOLD	(multi echo) GE/SE EPI or asymmetric SE EPI	DBV OEF (CMRO_2_)	endogenous	requires additional field map

## Data Availability

All data is available upon request.

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
