# Peer review of "Non-Invasive Evaluation of Cerebral Microvasculature Using Pre-Clinical MRI: Principles, Advantages and Limitations"

_diagnostics, 2021, doi:10.3390/diagnostics11060926_

Round 1
Reviewer 1 Report
The manuscript by Callewaert and colleagues provides an original comprehensive review of the available MRI techniques for assessment of the microcirculation, with a main focus on the cerebral microvascular bed. The manuscript covers both the principles of each technique and associated analysis strategies, as well as their main advantage and limitations. The manuscript is well organized and well written, presenting a vast amount of information in a simple way and not falling into over-technicalities.
Despite the merit of the authors, I would like to address the following minor comments/suggestions:
Given the numerous references of MRI techniques and analysis strategies to assess cerebral microcirculation throughout the text, I suggest that the title of the manuscript be changed to “Non-invasive evaluation of cerebral microvasculature using pre-clinical MRI: principles, advantages and limitations”.
In the Introduction section, the authors mention the importance of microcirculatory dysfunction in the pathophysiology of neurodegenerative diseases. I suggest that the authors add to the text that this dysfunction is currently though to be implicated in other neurologic (i.e., stroke) and psychiatric (i.e., schizophrenia) disorders.
There is an inconsistency regarding the writing of the terms “microvascular” and “microvasculature” – in a few instances the terms are incorrectly written as “micro-vascular” and “micro-vasculature”.
There is an inconsistency on the use of abbreviations – some are defined multiple times (e.g., EES, OEF); some are defined in the body of text and figure legends, whereas others are defined only in the text; some are not defined at all (e.g., MR – line 142; ROI – line 704) whereas some are defined despite being used only once (e.g., AUC). Given the high number of technical terms, the authors should choose carefully which terms to abbreviate in order not to compromise readability. Also, in formal written English, abbreviations are not used at the beginning of a sentence
Line 8: in affiliation #3 it reads “Nederland”. Shouldn’t it be “The Netherlands”?
Lines 44-48: This paragraph seems somewhat disconnected from the previous one. The authors should improve this paragraph by stating, for example, that endothelial dysfunction plays a central role in microvascular dysfunction. In addition, the authors should add that endothelial dysfunction also favors the development of thrombosis.
Lines 52 and 58 repeat the information obtained by MRI. It would be better to change/remove the information in one of the lines.
The text in lines 49 and 54-55 appears to provide contradictory information. In line 49 the authors state that microvessels are below the currently available MRI spatial resolution. However, in lines 54-55 they state that MRI has become one of the best suited techniques to assess microcirculation and perfusion. The authors should better state which features of the microcirculation can be assessed with MRI and which cannot.
The authors should provide more references for the text of the paragraph between lines 61-74 regarding clinical and pre-clinical studies.
Line 66: Instead of “represents” it should read “represent”
Lines 113-114: it would improve textual clarity if the authors stated, even if briefly, why “temperature, breathing rate and cardiac cycle of the animal under anesthesia is required”. Is it simply to guarantee the respiratory and hemodynamic stability of the anesthetized animal and/or is it to improve image acquisition?
Lines 114-115: Instead of “cardiac rate” it should read “heart rate”
Line 141: Even though
Lines 147-148: The two sentences can be merged together, like “There are two main approaches to quantitatively analyze DCE measurements - parametric and nonparametric approaches”.
Line 167: Instead of “straight forward” it should read “straightforward”
Line 180: Instead of “whit” it should read “with”
Line 182: Instead of “wildly” it should read “widely”
Lines 205 and 573: the comma after “Since” should be removed
Line 238: instead of “ration” it should read “ratio”
Line 247: instead of “(ml/g).” it should read “(ml/g);”
Lines 247-248: considering that the units of “CBV” and “CBF”, the authors should mention the units of “MTT”
Line 295: The comma after “The second” should be removed
Lines 235, 351: Keeping with the British English style of the text, instead of “labeling” it should read “labelling”
Line 352: The comma after “Even though” should be removed
Lines 367 and 569: Because the authors begin this sentence with “On the other hand,”, they should have started a previous sentence with “On the one hand,”
Line 369: Instead of “stregnth” it should read “strength”
Line 381: “pCO2” should be written as “pCO2”
Lines 398-399: The “bold” quality of the text should be removed
Line 408: Instead of “consist” it should read “consists”
Line 435: Instead of “describs” it should read “describes”
Line 441: Keeping with the British English style of the text, instead of “behavior” it should read “behaviour”
Line 490: Instead of “conttrast” it should read “contrast”
Line 552: Instead of “extra vascular” it should read “extravascular”
Line 598: Instead of “TI” it should read “T1”
Line 619: Instead of “half life time” shouldn’t it read “half-life”?
Line 678: Instead of “studies. Especially” it should read “studies, especially”
Line 721: The comma after “This” should be removed
Line 729: The definition of “SvO2” is not correct – it refers to “mixed venous oxygen saturation”
Author Response
Dear Reviewer,
Thank you for giving us the opportunity to submit a revised draft of the manuscript. We appreciate the time and effort that you have dedicated to providing valuable feedback on our manuscript. We are grateful to the reviewer for the insightful comments. We have incorporated changes to reflect the suggestions provided by the reviewers. We have highlighted the changes within the manuscript using the Word track changes feature. Furthermore, we have attached a point-by-point response letter containing our responses to the reviewers comments and concerns. The line numbers in the response letter refer to the revised manuscript file with the track changes feature enable.
Sincerely,
Bram Callewaert
Doctoral Researcher
KU Leuven University
Email: bram.callewaert@kuleuven.be

Reviewer 2 Report
This article may be more informative with an update of phase-contrast MRI in this field.
Otherwise, the article is well written.
Author Response
Dear Reviewer,
Thank you for giving us the opportunity to submit a revised draft of the manuscript. We appreciate the time and effort that you have dedicated to providing valuable feedback on our manuscript. Based on the reviewer's suggestion we incorporated a chapter about phase-contrast MRI. Please find attached the updated version of the manuscript. We have highlighted the changes within the manuscript using the Word track changes feature.
Sincerely,
Bram Callewaert
Doctoral Researcher
KU Leuven University
mail: bram.callewaert@kuleuven.be

Reviewer 3 Report
see the attached file

Author Response
Dear Reviewer,
Thank you for giving us the opportunity to submit a revised draft of the manuscript. We appreciate the time and effort that you have dedicated to providing valuable feedback on our manuscript. We are grateful to the reviewer for the insightful comments. We have incorporated changes to reflect the suggestions provided by the reviewers. We have highlighted the changes within the manuscript using the Word track changes feature. Furthermore, we have attached a point-by-point response letter containing our responses to the reviewers comments and concerns. The line numbers in the response letter refer to the revised manuscript file with the track changes feature enable.
Sincerely,
Bram Callewaert
Doctoral Researcher
KU Leuven University
mail: bram.callewaert@kuleuven.be

Round 2
Reviewer 3 Report
The revised manuscript reads much better and I do not see any essential issues. A few minor ones can be left at the authors' disposal without the need in another review round.
1. L695: "In order to quantitatively measure the VSI from the ΔR2*/ΔR2 other parameters such as the water diffusion rate, D, the contrast agent concentration, and the absolute blood volume fraction (CBV) should be considered" -- This can be confused with using the ratio ΔR2*/ΔR2, as in the steady-state VSI. Suggest to change for "ΔR2*and ΔR2".
2. L706: "Since, the absolute CBV cannot be measured, the steady-state approach requires an invasive measurement of the blood contrast agent concentration" -- The connection between the two statements is obscure. The non-measurable CBV cancels in the ratio, which is the reason for using this form of data processing.
3. Analogously, the non-measurable contrast concentration in the dynamic VSI cancels in the ratio ΔR2/(ΔR2*)^(3/2). The CBV is then measured as in DCS MRI. I would suggest to write the descriptions of the two VSI versions in a more "parallel" way.
4. L769: "Recently, a combined SAGE..." -- This method was de facto used already in 2005 by Kiselev et al., of course without giving this name proposed later by Schmiedeskamp et al. and extended to DSC MRI.
5. "Phase Contrast MRI" -- The title of this section can be confused with the phase contrast MRI used for SWI and QSM. Suggest to rename to, e.g., "Phase Contrast Flow MRI".
6. L687: "The dimensionless ratio ΔR2*/ΔR2, which increases with increases in the vessel diameter increase" -- this sentence might need revision.
7. In general, there is no separation between the biophysical cores of different methods (e.g., the susceptibility effect of contrast agent, phase contrast, diffusion effect and so on) and the measurement techniques (multi-echo acquisition, multi-band excitation, EPI etc.). It might be helpful to see a cross-table summarizing the biophysics vs. the measurement techniques. This is just a feeling, not a real suggestion, surely left to the authors' disposal.
Author Response
Dear Reviewer,
Thank you for giving us the opportunity to submit a revised draft of the manuscript. We appreciate the time and effort that you have dedicated to providing valuable feedback on our manuscript. We are grateful for the insightful comments. We have incorporated changes to reflect the provided suggestions. We have highlighted the changes within the manuscript using the Word track changes feature. Furthermore, we have attached a point-by-point response letter containing our responses to the reviewer’s comments.
Sincerely,
Bram Callewaert
